# A Novel Integrated Interval Rough MCDM Model for Ranking and Selection of Asphalt Production Plants

Bojan Matić [1], Stanislav Jovanović [1], Milan Marinković [1], Siniša Sremac [1], Dillip Kumar Das [2] and Željko Stević [3,*]

1 Faculty of Technical Sciences, University of Novi Sad, Trg Dositeja Obradovića 6, 21 000 Novi Sad, Serbia; bojanm@uns.ac.rs (B.M.); stasha@uns.ac.rs (S.J.); milan.marinkovic@uns.ac.rs (M.M.); sremacs@uns.ac.rs (S.S.)
2 Civil Engineering, Sustainable Transportation Research Group, School of Engineering, University of Kwazulu Natal, 238 Mazisi Kunene Rd, Glenwood, Durban 4041, South Africa; dasd@ukzn.ac.za
3 Faculty of Transport and Traffic Engineering, University of East Sarajevo, Vojvode Mišića 52, 74000 Doboj, Bosnia and Herzegovina
* Correspondence: zeljko.stevic@sf.ues.rs.ba

**Abstract:** Asphalt production plants play an important role in the field of civil engineering, but also in the entire economic system since the construction of roads enables uninterrupted functioning within it. In this paper, the ranking of asphalt production plants on the territory of the Autonomous Province of Vojvodina has been performed. The modern economy needs contemporary models and methods to solve complicated MCDM problems and, for these purposes, it has been developed an original Interval Rough Number (IRN) Multi-criteria decision-making (MCDM) model that implies an extension of two methods belonging to the field with interval rough numbers. After forming a list of eight most significant criteria for assessing the efficiency of asphalt production plants, the Interval Rough Number PIvot Pairwise RElative Criteria Importance Assessment (IRN PIPRECIA) method was developed to determine the significance of the criteria. A total of 21 locations with asphalt mixture installation were considered. For that purpose, seven asphalt production plants were included, and for their ranking, the IRN EDAS (Evaluation based on Distance from Average Solution) method was created. The aim of this paper is to develop a novel interval rough model that can be useful for determining the efficiency of asphalt production plants. Averaging in group decision-making (GDM) for both methods was performed using an IRN Dombi weighted geometric averaging (IRNDWGA) aggregator. The obtained results show that (A15) Ruma (SP)–Mačvanska Mitrovica–Zasavica has the best characteristics out of the set of locations considered in this study. However, Alternatives A6 and A19 are also variants with remarkably good characteristics since there is very little difference in values compared to the first-ranked alternative. Also, the obtained results have shown that the developed model is applicable, which is proven through a comparative analysis.

**Keywords:** asphalt production plants; IRN PIPRECIA; IRN EDAS; road construction

## 1. Introduction

In today's modern environment conditions in which we operate, transportation and logistics networks play a very important role in adequately achieving the sustainability of the economic system. Their task is to enable the movement of all types of flows, fostering further development and searching for optimal solutions within the processes that are being implemented. In order to be able to do that, appropriate traffic infrastructure, which primarily refers to roads as the most common way of transportation, is necessary. Therefore, it is necessary to constantly manage the area of road construction, as well as to take into account the type of roads, its construction and the lifespan that is causally related to its maintenance. Road construction and maintenance have a great impact on the environment, so, according to Sollazo et al. [1], asphalt production technology needs to be properly researched in order to quantify the environmental impact. In order to enable the fastest

and most efficient construction of roads, there are a large number of asphalt production plants that serve a certain narrow geographical area. Since, on the one hand, we strive to rationalize costs and resources in asphalt production, and, on the other hand, extend the lifespan of the road as long as possible, it is necessary to evaluate the efficiency of asphalt production plants. For this purpose, it is possible to apply various scientific methods that, according to Pasha et al. [2], enable resource savings in road construction management. If we take into account that the asphalt plant is a complex system that has goals to produce a high quality hot asphalt mixture, the assessment of their efficiency is of greater importance. The role in creating the efficiency of these plants is played by their technological and structural features, which depend on the method of the production of asphalt mass. Due to the possibility of achieving energy savings, greater utilization of resources and the possibility of meeting increasingly demanding standards in the asphalt industry, asphalt production plants are being upgraded with various additional systems. Recently, one of the most popular upgrades is the system for the use of foamed bitumen (Schaumbitumen) for low-temperature asphalt production, which is the future in the asphalt industry.

In the already mentioned research, Pasha et al. [2] has applied a fuzzy MCDM model that consists of a combination of DEMATEL (decision-making trial and evaluation laboratory), Fuzzy ANP (analytical network process) and Fuzzy TOPSIS (technique for order preference by similarity to ideal solution). The research and application of the model was focused on the selection of road pavement type. Three MCDM methods: Weighted Aggregate Sum Product Assessment (WASPAS), TOPSIS and Fuzzy AHP, were used in [3] to evaluate the effect of various types of fibers on the mechanical performance of bituminous mixtures. In their research, Chao et al. [4] applied an analytic hierarchy process (AHP) as a quantitative approach for recycling method selection depending on road performance and construction costs. The combination of fuzzy AHP and Fuzzy TOPSIS method was applied in [5] for the subjective analysis of the pavement conditions for automated maintenance prioritization. The results obtained using the integrated AHP–VIKOR (VIšeKriterijumska Optimizacija i Kompromisno Rešenje) model in the research [6] indicate that hot mix asphalt with reclaimed asphalt pavement (HMAR) shows the best socio-economic characteristics, while hot mix asphalt with warm mix additive Sasobit (HMAW) has the best eco-friendly characteristics.

In addition to the professional contribution that is reflected in determining the efficiency through the ranking of asphalt production plants that can have an impact on the environment, but also the functioning of the economic system, the following scientific contributions are highlighted. Techniques belonging to the field of MCDM, especially in terms of integration with rough and interval rough numbers [7,8], represent an extremely powerful and useful approach for more precise decision-making in a group decision-making process. An extension of the PIPRECIA and EDAS methods with interval rough numbers has been performed, which is a great contribution of this study (IRN PIPRECIA–IRN EDAS model). A detailed algorithm of both developed approaches is presented in Section 3. The contribution can be seen through the development of an entire interval rough MCDM model, implying an extension of the PIPRECIA method, which determines the significance of the criteria, and the EDAS method, which is used to rank potential solutions. The new Interval Rough EDAS approach is used to assess the efficiency of asphalt plants through the locations of the roads they serve, which enables the reduction of subjectivity and imprecisions that occur on a daily basis when making decisions. This model provides an objective aggregation of experts' decisions with full respect for inaccuracy and subjectivity that prevails in group decision-making.

After the introductory section of the paper, the rest is structured as follows. Section 2 presents a description of the preliminaries necessary to carry out the development of a novel IRN MCDM model. Section 3 involves the development of the novel IRN MCDM model with explanations for each step individually. In Section 4, a comparative analysis of the obtained results with three other IRN MCDM methods is performed. Lastly, in

Section 5, a brief overview of the most important novelties and contributions of the paper is provided, as well as a proposal for the continuation of the research.

## 2. Preliminaries

In this section of the paper, the main settings of the theory of interval rough numbers and main operations with them are presented in order to be able to create new interval rough models later in the paper.

IRNs are described by specific arithmetic operations that are varied from arithmetic operations with classical RNs. Arithmetic operations between two interval rough numbers $IRN(A) = ([a_1, a_2], [a_3, a_4])$ and $IRN(B) = ([b_1, b_2], [b_3, b_4])$ are performed using Equations (1)–(5) [9,10]:

(1) Addition of interval rough numbers, "+",

$$IRN(A) + IRN(B) = ([a_1, a_2], [a_3, a_4]) + ([b_1, b_2], [b_3, b_4]) = ([a_1 + b_1, a_2 + b_2], [a_3 + b_3, a_4 + b_4]) \tag{1}$$

(2) Subtraction of interval rough numbers, "−",

$$IRN(A) - IRN(B) = ([a_1, a_2], [a_3, a_4]) - ([b_1, b_2], [b_3, b_4]) = ([a_1 - b_4, a_2 - b_3], [a_3 - b_2, a_4 - b_1]) \tag{2}$$

(3) Multiplication of interval rough numbers, "×",

$$IRN(A) \times IRN(B) = ([a_1, a_2], [a_3, a_4]) \times ([b_1, b_2], [b_3, b_4]) = ([a_1 \times b_1, a_2 \times b_2], [a_3 \times b_3, a_4 \times b_4]) \tag{3}$$

(4) Division of interval rough numbers, "/",

$$IRN(A) / IRN(B) = ([a_1, a_2], [a_3, a_4]) / ([b_1, b_2], [b_3, b_4]) = ([a_1 / b_4, a_2 / b_3], [a_3 / b_2, a_4 / b_1]) \tag{4}$$

(5) Scalar multiplication of interval rough numbers where $k > 0$

$$k \times IRN(A) = k \times ([a_1, a_2], [a_3, a_4]) = ([k \times a_1, k \times a_2], [k \times a_3, k \times a_4]) \tag{5}$$

Any two interval rough numbers $IRN(\alpha) = \left([\alpha^L, \alpha^U], [\alpha'^L, \alpha'^U]\right)$ and $IRN(\beta) = \left([\beta^L, \beta^U], [\beta'^L, \beta'^U]\right)$ are ranked according to Figure 1.

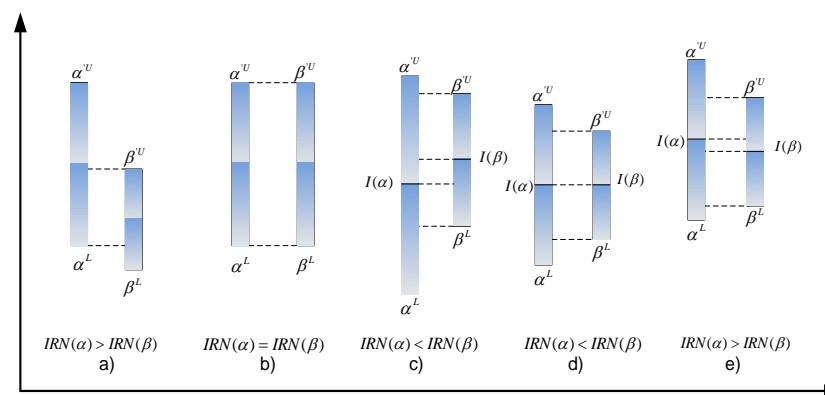

**Figure 1.** Ranking interval rough numbers.

The intersection points of IRNs are obtained as follow:

$$\mu_\alpha = \frac{RB(\alpha_{ui})}{RB(\alpha_{ui}) + RB(\alpha_{li})}; RB(\alpha_{ui}) = \alpha'^U - \alpha'^L; RB(\alpha_{li}) = \alpha^U - \alpha^L \tag{6}$$

$$\mu_\beta = \frac{RB(\beta_{ui})}{RB(\beta_{ui}) + RB(\beta_{li})}; RB(\beta_{ui}) = \beta'^U - \beta'^L; RB(\beta_{li}) = \beta^U - \beta^L \tag{7}$$

$$I(\alpha) = \mu_\alpha \cdot \alpha^L + (1 - \mu_\alpha) \cdot \alpha'^U \tag{8}$$

$$I(\beta) = \mu_\beta \cdot \beta^L + (1 - \mu_\beta) \cdot \beta'^U \tag{9}$$

## 3. Novel Interval Rough MCDM Models

This section of the paper presents new MCDM models applied to evaluate the efficiency of asphalt production plants in the Autonomous Province of Vojvodina. After main operations with interval rough numbers previously presented in order to facilitate understanding and further following the paper, an overview of the newly developed interval rough models is given. The PIPRECIA and EDAS methods have been extended with interval rough numbers, which is a great contribution of the study (IRN PIPRECIA–IRN EDAS). A detailed algorithm of both developed approaches is shown below. The contribution can be seen through the development of an entire interval rough MCDM model, implying an extension of the PIPRECIA method, which determines the significance of criteria, and the EDAS method, which is used to rank potential solutions.

### 3.1. A Novel Interval Rough PIPRECIA Method

In a very short time since appearing, the PIPRECIA method [11] has found its application in various fields [12,13], so it has become an interesting method for researchers to determine the weight values of criteria. Proof of this is given by its extensions: Fuzzy PIPRECIA [14], which has been quite exploited [15–18] due to the treatment of problems including fuzzy logic. In addition, an extension of this method can be found with grey theory for Personnel Selection [19]. As highlighted in this section, this method has been extended with interval rough numbers, and the steps of the proposed approach are as follows.

Step 1. Since the methodology with interval rough numbers differs in relation to crisp theory, it is first necessary to create linguistic scales for assessing the significance of criteria, i.e., their mutual comparison. Due to the nature of the method itself and the theory of rough sets, it is necessary to develop two linguistic scales that are transformed into interval rough numbers. The first linguistic scale (Table 1) refers to situations when the $C_j$ criterion is more significant compared to the previous $C_{j-1}$ criterion.

**Table 1.** Criterion evaluation by linguistic scale 1-2.

| Linguistic Term | Abbr. | | | Interval Rough Number | |
|---|---|---|---|---|---|
| Almost equal value | AE | | 1 | [1.00, 1.05] | [1.10, 1.10] |
| Slightly more significant | SM | | 2 | [1.10, 1.20] | [1.20, 1.25] |
| Moderately more significant | MMS | | 3 | [1.20, 1.35] | [1.30, 1.40] |
| More significant | M | Scale 1-2 | 4 | [1.30, 1.50] | [1.40, 1.55] |
| Much more significant | MM | | 5 | [1.40, 1.65] | [1.50, 1.70] |
| Dominantly more significant | DM | | 6 | [1.50, 1.80] | [1.60, 1.85] |
| Absolutely more significant | AM | | 7 | [1.60, 1.90] | [1.70, 1.95] |

The second linguistic scale (Table 2) refers to situations when the $C_j$ criterion is less significant comparing to the previous $C_{j-1}$ criterion.

**Table 2.** Criterion evaluation by linguistic scale 0-1.

| Linguistic Term | Abbr. | | | Interval Rough Number | |
|---|---|---|---|---|---|
| Weakly less significant | WL | | 1 | [0.80, 0.90] | [0.85, 0.95] |
| Moderately less significant | MLS | | 1/2 | [0.70, 0.80] | [0.75, 0.85] |
| Less significant | L | Scale 0-1 | 1/3 | [0.60, 0.70] | [0.65, 0.75] |
| Really less significant | RL | | 1/4 | [0.50, 0.60] | [0.55, 0.65] |
| Much less significant | ML | | 1/5 | [0.40, 0.50] | [0.45, 0.55] |
| Dominantly less significant | DL | | 1/6 | [0.30, 0.40] | [0.35, 0.45] |
| Absolutely less significant | AL | | 1/7 | [0.20, 0.30] | [0.25, 0.35] |

Step 2. This step involves defining all the elements for forming the MCDM model. Since the precondition for the application of such a methodology is group decision-making, it is necessary to define a group of decision-makers (DMs) in addition to the criteria. Further,

it is important to note that the Interval Rough PIPRECIA procedure does not require any sorting of criteria before their evaluation by DMs.

Step 3. In this step, the significance of the criteria is assessed by each DM. When evaluating the significance of the criteria, one starts from the second criterion and evaluates the significance in relation to the previous criterion using one of the above scales. It is necessary to apply Equation (10) and perform an evaluation depending on whether the criterion is more or less significant comparing to the previous one.

$$IRN\left[s_j^r\right] = \begin{cases} > [1,1],[1,1] \ if \ C_j > C_{j-1} \\ = [1,1],[1,1] \ if \ C_j = C_{j-1} \\ < [1,1],[1,1] \ if \ C_j < C_{j-1} \end{cases} \tag{10}$$

$IRN\left[s_j^r\right]$ represents a comparison of the criteria by each decision-maker $r$.

Since this is a group decision-making, as already emphasized, it is necessary to apply some of the operators for averaging the values of all DMs, in order to obtain a single initial interval rough decision-making matrix. In this paper, averaging is performed using an IRN Dombi weighted geometric averaging (IRNDWGA) aggregator, Equation (11), based on the research carried out in [20].

$$IRNDWGA\{IRN(\varphi_1), IRN(\varphi_2), \dots, IRN(\varphi_n)\} = \left( \begin{bmatrix} \dfrac{\Sigma_{j=1}^n \varphi_{lj}}{1+\left\{\sum\limits_{j=1}^n w_j \left(\dfrac{1-f\left(\varphi_{lj}\right)}{f\left(\varphi_{lj}\right)}\right)^\rho\right\}^{1/\rho}}, \dfrac{\Sigma_{j=1}^n \overline{\varphi}_{lj}}{1+\left\{\sum\limits_{j=1}^n w_j \left(\dfrac{1-f\left(\overline{\varphi}_{lj}\right)}{f\left(\overline{\varphi}_{lj}\right)}\right)^\rho\right\}^{1/\rho}} \\ \dfrac{\Sigma_{j=1}^n \varphi_{uj}}{1+\left\{\sum\limits_{j=1}^n w_j \left(\dfrac{1-f\left(\varphi_{uj}\right)}{f\left(\varphi_{uj}\right)}\right)^\rho\right\}^{1/\rho}}, \dfrac{\Sigma_{j=1}^n \overline{\varphi}_{uj}}{1+\left\{\sum\limits_{j=1}^n w_j \left(\dfrac{1-f\left(\overline{\varphi}_{uj}\right)}{f\left(\overline{\varphi}_{uj}\right)}\right)^\rho\right\}^{1/\rho}} \end{bmatrix} \right) \tag{11}$$

Step 4. Calculation of the coefficient $IRN\left[k_j\right]$.

$$IRN\left[k_j\right] = \begin{cases} = [1,1],[1,1] \ if \ j = 1 \\ 2 - [s_j] \quad if \ j > 1 \end{cases} \tag{12}$$

Step 5. Calculation of the interval rough weight $IRN\left[q_j\right]$

$$RN\left[q_j\right] = \begin{cases} = [1,1],[1,1] \ if \ j = 1 \\ \dfrac{[q_{j-1}]}{[k_j]} \quad if \ j > 1 \end{cases} \tag{13}$$

Step 6. Calculation of the relative interval rough weight of the criterion $IRN\left[w_j\right]$

$$IRN\left[w_j\right] = \dfrac{[q_j]}{\sum\limits_{j=1}^n [q_j]} \tag{14}$$

The following steps describe the application of the inverse Interval Rough PIPRECIA method which is an integral part of the procedure.

Step 7. Evaluation of the comparative significance of the criteria by each DM, but in a way that it starts from the penultimate criterion, as follows.

$$
IRN\left[s_j^{r\prime}\right] = \begin{cases} > [1,1],[1,1] \; if \; C_j > C_{j+1} \\ = [1,1],[1,1] \; if \; C_j = C_{j+1} \\ < [1,1],[1,1] \; if \; C_j < C_{j+1} \end{cases} \tag{15}
$$

$IRN\left[s_j^{r\prime}\right]$ denotes the assessment of criteria by a decision-maker *r*.

Again, it is required to specify one of the averaging operators.

Step 8. Calculation of the coefficient $IRN\left[k_j{}'\right]$

$$
IRN\left[k_j{}'\right] = \begin{cases} = [1,1],[1,1] \; if \; j = n \\ 2 - \left[s_j{}'\right] \; if \; j > n \end{cases} \tag{16}
$$

Step 9. Calculation of the interval rough weight $IRN\left[q_j{}'\right]$

$$
\overline{q_j{}'} = \begin{cases} = [1,1],[1.1] \; if \; j = n \\ \dfrac{\overline{q_{j+1}{}'}}{\overline{k_j{}'}} \; if \; j > n \end{cases} \tag{17}
$$

Step 10. Determining the relative interval rough weight of the criterion $IRN\left[w_j{}'\right]$

$$
IRN\left[w_j{}'\right] = \frac{\left[q_j{}'\right]}{\sum\limits_{j=1}^{n}\left[q_j{}'\right]} \tag{18}
$$

Step 11. In order to obtain the final values of the criteria, it is necessary to apply the following equation.

$$
\left[w_j{}''\right] = \frac{1}{2}\left(IRN\left[w_j\right] + IRN\left[w_j{}'\right]\right) \tag{19}
$$

Step 12. Calculation of Spearman and Pearson correlation coefficients.

*3.2. A novel Interval Rough EDAS*

As an important contribution made in the paper is also the development of a novel Interval Rough EDAS approach to assess the efficiency of asphalt bases that enables to reduce the subjectivity and imprecisions that occur on a daily basis in decision-making. The newly developed interval Rough EDAS model represents a contribution to the literature of multi-criteria decision-making, providing an objective aggregation of experts' decisions with full respect for inaccuracy and subjectivity that prevails in group decision-making.

The EDAS [21] method belongs to the group of newer methods of multi-criteria decision-making. In a short time, it has found wide application in solving both engineering problems and problems in the field of business decision-making. This method has a large number of extensions: fuzzy EDAS [22], interval grey EDAS [23], picture fuzzy EDAS [24], Rough EDAS [25], Interval-valued Pythagorean Fuzzy EDAS method [26], etc. This approach can represent important support during a decision-making process in everyday situations where conflicts when considering parameters are frequent.

Step 1: After defining input parameters, which include defining *m* alternatives and *n* criteria, it is necessary to model a group decision-making process by forming a group of *k* experts who will evaluate the alternatives, considering each criterion individually.

Step 2. Transformation of individual decision matrices into a group interval rough matrix shown by Equation (20):

$$IRN(X_{ij}) = \begin{array}{c} \\ A_1 \\ A_2 \\ \cdots \\ A_m \end{array} \begin{array}{c} C_1 \quad C_2 \quad \ldots \quad C_n \\ \left[ \begin{array}{cccc} IRN(x_{11}) & IRN(x_{12}) & \ldots & IRN(x_{1n}) \\ IRN(x_{21}) & IRN(x_{22}) & & IRN(x_{2n}) \\ \cdots & \cdots & \cdots & \cdots \\ IRN(x_{l1}) & IRN(x_{l2}) & \ldots & IRN(x_{ln}) \end{array} \right]_{m \times n} \end{array} \tag{20}$$

Step 3. In this step, it is necessary to calculate an average solution by forming an $IRN(AV_j)$ matrix (21)

$$IRN(AV_j) = \left[ av_j^L, av_j^U \right], \left[ av_j^{L\prime}, av_j^{U\prime} \right]_{1 \times n} \tag{21}$$

by applying Equation (22)

$$\sum_{i=1}^{m} \frac{IRN(x_{ij})}{m} = \sum_{i=1}^{m} \left[ \frac{IRN\left(x_{ij}^L\right)}{m}, \frac{IRN\left(x_{ij}^U\right)}{m} \right], \sum_{i=1}^{m} \left[ \frac{IRN\left(x_{ij}^{L\prime}\right)}{m}, \frac{IRN\left(x_{ij}^{U\prime}\right)}{m} \right] \tag{22}$$

Step 4. Calculation of positive distance *IRN(PDAij)* (23) and negative distance *IRN(NDAij)* (24) in relation to the average solution *IRN(AVj)* for all criteria.

$$IRN(PDA_{ij}) = \left[ pda_{ij}^L, pda_{ij}^U \right], \left[ pda_{ij}^{L\prime}, pda_{ij}^{U\prime} \right]_{m \times n} \tag{23}$$

$$IRN(NDA_{ij}) = \left[ nda_{ij}^L, nda_{ij}^U \right], \left[ nda_{ij}^{L\prime}, nda_{ij}^{U\prime} \right]_{m \times n} \tag{24}$$

In order to obtain the elements of these matrices, it is necessary to take into account the type of criteria, so if it is necessary to be maximized, i.e., if it belongs to the benefit group, Equations (25) and (26) are applied:

$$IRN\left(PDA_{ij}\right) = \left[ pda_{ij}^L, pda_{ij}^U \right], \left[ pda_{ij}^{L\prime}, pda_{ij}^{U\prime} \right] = \left[ \frac{b_{ij}^L}{av_j^{U\prime}}, \frac{b_{ij}^U}{av_j^{L\prime}} \right], \left[ \frac{b_{ij}^{L\prime}}{av_j^U}, \frac{b_{ij}^{U\prime}}{av_j^L} \right]$$
$$IRN\left(B_{ij}\right) = \left[ b_{ij}^L, b_{ij}^U \right], \left[ b_{ij}^{L\prime}, b_{ij}^{U\prime} \right] = \max\left( 0, \left[ x_{ij}^L - av_j^{U\prime}, x_{ij}^U - av_j^{L\prime} \right], \left[ x_{ij}^{L\prime} - av_j^U, x_{ij}^{U\prime} - av_j^L \right] \right) \tag{25}$$

$$IRN\left(NDA_{ij}\right) = \left[ nda_{ij}^L, nda_{ij}^U \right], \left[ nda_{ij}^{L\prime}, nda_{ij}^{U\prime} \right] = \left[ \frac{b_{ij}^L}{av_j^{U\prime}}, \frac{b_{ij}^U}{av_j^{L\prime}} \right], \left[ \frac{b_{ij}^{L\prime}}{av_j^U}, \frac{b_{ij}^{U\prime}}{av_j^L} \right]$$
$$IRN\left(B_{ij}\right) = \left[ b_{ij}^L, b_{ij}^U \right], \left[ b_{ij}^{L\prime}, b_{ij}^{U\prime} \right] = \max\left( 0, \left[ av_j^L - x_{ij}^{U\prime}, av_j^U - x_{ij}^{L\prime} \right], \left[ av_j^{L\prime} - x_{ij}^U, av_j^{U\prime} - x_{ij}^L \right] \right) \tag{26}$$

As can be seen, the difference is in the calculation of *IRN(Bij)*.

If the criterion needs to be minimized, i.e., if it belongs to the cost group of criteria, Equations (27) and (28) are applied.

$$IRN\left(PDA_{ij}\right) = \left[ pda_{ij}^L, pda_{ij}^U \right], \left[ pda_{ij}^{L\prime}, pda_{ij}^{U\prime} \right] = \left[ \frac{b_{ij}^L}{av_j^{U\prime}}, \frac{b_{ij}^U}{av_j^{L\prime}} \right], \left[ \frac{b_{ij}^{L\prime}}{av_j^U}, \frac{b_{ij}^{U\prime}}{av_j^L} \right]$$
$$IRN\left(B_{ij}\right) = \left[ b_{ij}^L, b_{ij}^U \right], \left[ b_{ij}^{L\prime}, b_{ij}^{U\prime} \right] = \max\left( 0, \left[ av_j^L - x_{ij}^{U\prime}, av_j^U - x_{ij}^{L\prime} \right], \left[ av_j^{L\prime} - x_{ij}^U, av_j^{U\prime} - x_{ij}^L \right] \right) \tag{27}$$

$$IRN\left(NDA_{ij}\right) = \left[ nda_{ij}^L, nda_{ij}^U \right], \left[ nda_{ij}^{L\prime}, nda_{ij}^{U\prime} \right] = \left[ \frac{b_{ij}^L}{av_j^{U\prime}}, \frac{b_{ij}^U}{av_j^{L\prime}} \right], \left[ \frac{b_{ij}^{L\prime}}{av_j^U}, \frac{b_{ij}^{U\prime}}{av_j^L} \right]$$
$$IRN\left(B_{ij}\right) = \left[ b_{ij}^L, b_{ij}^U \right], \left[ b_{ij}^{L\prime}, b_{ij}^{U\prime} \right] = \max\left( 0, \left[ x_{ij}^L - av_j^{U\prime}, x_{ij}^U - av_j^{L\prime} \right], \left[ x_{ij}^{L\prime} - av_j^U, x_{ij}^{U\prime} - av_j^L \right] \right) \tag{28}$$

The formulas previously created require additional constraints due to the nature of interval rough numbers. Precisely due to the nature of the IRN, i.e., the existence of the lower and upper limit, the following situations related to the *IRN(Bij)* calculation is possible:

(1) if the sum of the lower and upper limits of the interval rough number is less than zero, then Equation (29) is applied.

$$if \left[b_{ij}^{L} + b_{ij}^{U}\right], \left[b_{ij}^{L\prime} + b_{ij}^{U\prime}\right] \leq 0 \ then \ IRN(B_{ij}) = 0 \tag{29}$$

(2) if the sum of the lower and upper limits is greater than zero and each element of the IRN is greater than zero, then the value of that IRN number is retained in the *IRN(Bij)* as shown by Equation (30).

$$if \left[b_{ij}^{L} + b_{ij}^{U}\right], \left[b_{ij}^{L\prime} + b_{ij}^{U\prime}\right] > 0 \ and \ if \left[b_{ij}^{L} > 0, b_{ij}^{U} > 0\right], \left[b_{ij}^{L\prime} > 0, b_{ij}^{U\prime} > 0\right] then IRN(B_{ij}) = IRN(B_{ij}) \tag{30}$$

(3) if the sum of the lower and upper limits is greater than zero, and the lower limit of IRN is less than zero, then the absolute value of that number is taken, i.e., *IRN(Bij)*, Equation (31).

$$if \left[b_{ij}^{L} + b_{ij}^{U}\right], \left[b_{ij}^{L\prime} + b_{ij}^{U\prime}\right] > 0 \ and \ if \left[b_{ij}^{L} < 0, b_{ij}^{U} > 0\right], \left[b_{ij}^{L\prime} < 0, b_{ij}^{U\prime} > 0\right] then IRN(B_{ij}) = IRN|B_{ij}| \tag{31}$$

Step 5. Multiplying the interval rough matrix *IRN(PDAij)* and *IRN(NDAij)* by the weight values of the criteria, Equations (32) and (33).

Multiplying the interval rough matrix by the weight values of the criteria, Equations (32) and (33).

$$IRN(VP_{ij}) = \left[vp_{ij}^{L}, vp_{ij}^{U}\right], \left[vp_{ij}^{L\prime}, vp_{ij}^{U\prime}\right]_{m \times n} = \left[pda_{ij}^{L} \times w_{j}^{L}, pda_{ij}^{U} \times w_{j}^{U}\right], \left[pda_{ij}^{L\prime} \times w_{j}^{L\prime}, pda_{ij}^{U\prime} \times w_{j}^{U\prime}\right] \tag{32}$$

$$IRN(VN_{ij}) = \left[vn_{ij}^{L}, vn_{ij}^{U}\right], \left[vn_{ij}^{L\prime}, vn_{ij}^{U\prime}\right]_{m \times n} = \left[nda_{ij}^{L} \times w_{j}^{L}, nda_{ij}^{U} \times w_{j}^{U}\right], \left[nda_{ij}^{L\prime} \times w_{j}^{L\prime}, nda_{ij}^{U\prime} \times w_{j}^{U\prime}\right] \tag{33}$$

Step 6. Calculation of the sum of previously weighted rough interval matrices, Equations (34) and (35).

$$IRN(SP_i) = \left[sp_i^{L}, sp_i^{U}\right], \left[sp_i^{L\prime}, sp_i^{U\prime}\right] = \sum_{i=1}^{n} IRN\left(VP_{ij}\right) \tag{34}$$

$$IRN(SN_i) = \left[sn_i^{L}, sn_i^{U}\right], \left[sn_i^{L\prime}, sn_i^{U\prime}\right] = \sum_{i=1}^{n} IRN\left(VN_{ij}\right) \tag{35}$$

Step 7. Calculation of normalized values for matrices from the previous step for all alternatives, Equations (36) and (37).

$$IRN(NSP_i) = \left[nsp_i^{L}, nsp_i^{U}\right], \left[nsp_i^{L\prime}, nsp_i^{U\prime}\right] = \frac{IRN(SP_i)}{\max IRN(SP_i)} = \left[\frac{sp_i^{L}}{\max sp_i^{U\prime}}, \frac{sp_i^{U}}{\max sp_i^{L\prime}}\right], \left[\frac{sp_i^{L\prime}}{\max sp_i^{U}}, \frac{sp_i^{U\prime}}{\max sp_i^{L}}\right] \tag{36}$$

$$IRN(NSN_i) = \left[nsn_i^{L}, nsn_i^{U}\right], \left[nsn_i^{L\prime}, nsn_i^{U\prime}\right] = 1 - \frac{IRN(SN_i)}{\max IRN(SN_i)} = 1 - \left[\frac{sn_i^{L}}{\max sn_i^{U\prime}}, \frac{sn_i^{U}}{\max sn_i^{L\prime}}\right], \left[\frac{sn_i^{L\prime}}{\max sn_i^{U}}, \frac{sn_i^{U\prime}}{\max sn_i^{L}}\right] \tag{37}$$

Step 8. Calculation of the values of all alternatives *IRN(ASi)* and their ranking in descending order, Equation (38).

$$IRN(AS_i) = \left[as_i^{L}, as_i^{U}\right], \left[as_i^{L\prime}, as_i^{U\prime}\right] = \left[\frac{IRN(NSP_i) + IRN(NSN_i)}{2}\right] \tag{38}$$

## 4. Evaluating the Efficiency of Plants for the Production of Asphalt Bases in Vojvodina

This section presents the creation of the MCDM model which defines a total of 21 locations used for ranking seven asphalt production plants and eight criteria used for assessing their efficiency. Alternatives, i.e., locations for road infrastructure maintenance by asphalt production plants, are on the territory of the Autonomous Province of Vojvodina and they are presented in the model as follows. (A1) Rumenka–Budisava, (A2) Rumenka–Kać, (A3) Rumenka–Veternik, (A4) Bački Petrovac–Bač I, (A5) Bački Petrovac–Bač II, (A6) Bački Petrovac–Bačka Palanka, (A7) Srbobran–Kula, (A8) Srbobran–Zrenjanin, (A9) Srbobran–Novi Sad, (A10) Ruma (K)–Zabalj, (A11) Ruma (K)–Fruška Gora, (A12) Ruma (K)–Jamena,

(A13) Ruma (SP)–N. Banovci–St. Pazova, (A14) Ruma (SP)–Ruma–Pećinci, (A15) Ruma (SP)–Mačvanska Mitrovica–Zasavica, (A16) Subotica–highway 71–73 km direction to NS, (A17) Subotica–highway 81–83 km direction to NS, (A18) Subotica–Kula, (A19) Pančevo–Devojački bunar, (A20) Pančevo–Sakule–Baranda, (A21) Pančevo–Strža–Jasenovo. The road infrastructure maintenance at the given locations is performed by seven asphalt production plants in the following way. The first three locations, A1–A3, belong to the plant in Rumenka, A4–A6 to the plant in Bački Petrovac, A7–A9 to the plant in Srbobran. There are two asphalt production plants in the city of Ruma: A10–A12 belongs to the first, and A13–A15 belongs to the second. Locations for road infrastructure maintenance, A16–A18, belong to the asphalt production plant in Subotica. The last plant for the production of asphalt is located in Pančevo and the locations A19–A21 belong to it.

The criteria used for the efficiency assessment, i.e., ranking of the asphalt production plants are as follows: C1–distance of the asphalt plant from the asphalting site (km), C2–asphalt plant capacity (t/h). One of the key elements in the selection of asphalt plant is the capacity of the plant itself, which depends on its type [27,28], and, in this study, it ranges from 100 to 200 t/h. C3–rank of the road connecting the plant and the place of asphalting, C4–type of asphalt mixture produced, C5–condition of the road network transporting the mixture, C6–capacity of the vehicle for transporting asphalt mixture (t), C7–type of stone for asphalt mixture production, C8–bitumen manufacturer. The complete initial matrix used for the evaluation in group decision-making is shown in Table 3.

**Table 3.** Initial matrix as an input parameter for a group decision-making process.

| | C1 | C2 | C3 | C4 | C5 | C6 | C7 | C8 |
|---|---|---|---|---|---|---|---|---|
| A1 | 28 | 160 | Ib-class state r., IIa-class state r. | AB11 | satisfactory | 20 | limestone | NIS Petrol |
| A2 | 23 | 160 | Ib-class state r., IIa-class state r. | AB16 | satisfactory | 20 | limestone | NIS Petrol |
| A3 | 16 | 160 | Ib-class state r., IIa-class state r. | BNHS16 | satisfactory | 10 | limestone | NIS Petrol |
| A4 | 50 | 150 | IIa-class state r. | BNS 22 | bad | 25 | limestone Kovilovača | OMW |
| A5 | 50 | 150 | IIa-class state r. | AB 11 | bad | 25 | limestone Kovilovača | OMW |
| A6 | 22 | 150 | IIa-class state r. | BNS 22 | good | 25 | limestone Kovilovača | OMW |
| A7 | 30 | 200 | Ib-class state r. | AB 11s | good | 25 | eruptive stone, Velika Bisina Raška | Pančevo |
| A8 | 60 | 200 | Ib-class state r., IIa-class state r. | AB 11s | good | 25 | eruptive stone, Velika Bisina Raška | Pančevo |
| A9 | 35 | 200 | IIa-class state r. | BNS22sA | good | 25 | linestone Kovilovača Despotovac | Refinery Pančevo |
| A10 | 61 | 100 | Ib-class state r. | BNS 22s | good | 25 | crushed gravel | Pančevo |
| A11 | 26 | 100 | Ib-class state r., IIb-class state r. | BNHS 16 | good | 25 | crushed gravel | Pančevo |
| A12 | 81 | 100 | Ia-class state r., IIa-class state r. | BNS 32s | good | 25 | crushed gravel | Pančevo |
| A13 | 55 | 120t | Ib-class state r., IIa-class state r. | AB 16s(PmB) | satisfactory | 30 | eruptive stone | NIS-Pančevo |
| A14 | 15 | 120 | Ib-class state r., Ia-class state r. | AB 11s(PmB) | good | 30 | eruptive stone | NIS-Pančevo |

**Table 3.** *Cont.*

|  | **C1** | **C2** | **C3** | **C4** | **C5** | **C6** | **C7** | **C8** |
|---|---|---|---|---|---|---|---|---|
| A15 | 25 | 120 | Ib-class state r., Ia-class state r. | BNS 32sA | good | 30 | limestone | NIS-Pančevo |
| A16 | 20 | 200 | Ia-class state r., Ib-class state r., IIa-class state r. | AB 11s | good | 25 | eruptive stone, Velika Bisina Raška | Pančevo |
| A17 | 10 | 200 | Ia-class state r., Ib-class state r., IIa-class state r. | AB 11s | good | 25 | eruptive stone, Velika Bisina Raška | Pančevo |
| A18 | 25 | 200 | Ib-class state r., IIa-class state r. | BNS22sA | good | 25 | limestone Kovilovača Despotovac | Refinery Pačevo |
| A19 | 44 | 160 | Ib 10 state r. | BNS 22sA, AB 11s | satisfactory | 30 | limestone | NIS |
| A20 | 54 | 160 | Ib 10 state r. | BNS 22sA, AB 11, AB 11s, BNHS 16 | satisfactory | 30 | limestone | NIS |
| A21 | 76 | 160 | Ib 14 state r. | BNS 22sA, BNS 32sA, PMB 11 | satisfactory | 30 | limestone | NIS |

Some of the characteristics of the locations with the installation of asphalt mixture by seven plants for asphalt mixture production on the basis of the criteria shown in Table 3 are: the distance of the asphalt plant from the asphalting site, ranging 10–80 km; plant capacity is 100–200 t/h as already mentioned. The rank of the road refers to: state road of class I, state road of class II, local road, diversity in the structure of the asphalt mixture that is produced. The condition of the road network by which the mixture is transported is assessed as bad, satisfactory, good and excellent. In addition, the characteristics of the type of stone for the production of asphalt mixture and bitumen producers are presented.

*4.1. Determining Criteria Weights Using the Novel IRN PIPRECIA Method*

Table 4 presents the assessment of the significance of the criteria by three decision-makers who participated in group decision-making. The assessment was performed on the basis of the developed scales shown in Tables 1 and 2.

**Table 4.** Assessment of criteria by three decision-makers.

|  | **C1** | **C2** | **C3** | **C4** | **C5** | **C6** | **C7** | **C8** |
|---|---|---|---|---|---|---|---|---|
| DM1 |  | [0.80, 0.90] [0.85, 0.95] | [0.3, 0.4] [0.35, 0.45] | [1.3, 1.5] [1.4, 1.55] | [0.2, 0.3] [0.25, 0.35] | [1.1, 1.2] [1.2, 1.25] | [0.6, 0.7] [0.65, 0.75] | [1.1, 1.2] [1.2, 1.25] |
| DM2 |  | [1.00, 1.05] [1.10, 1.10] | [0.4, 0.5] [0.45, 0.55] | [1.3, 1.5] [1.4, 1.55] | [0.3, 0.4] [0.35, 0.45] | [1, 1.05] [1.1, 1.1] | [0.5, 0.6] [0.55, 0.65] | [1, 1.05] [1.1, 1.1] |
| DM3 |  | [0.80, 0.90] [0.85, 0.95] | [0.4, 0.5] [0.45, 0.55] | [1.3, 1.5] [1.4, 1.55] | [0.3, 0.4] [0.35, 0.45] | [1.1, 1.2] [1.2, 1.25] | [0.5, 0.6] [0.55, 0.65] | [1.1, 1.2] [1.2, 1.25] |
|  | **C8** | **C7** | **C6** | **C5** | **C4** | **C3** | **C2** | **C1** |
| DM1 | [0.8, 0.9] [0.85, 0.95] | [1.1, 1.2] [1.2, 1.25] | [0.7, 0.8] [0.75, 0.85] | [1.1, 1.2] [1.2, 1.25] | [0.2, 0.3] [0.25, 0.35] | [1.3, 1.5] [1.4, 1.55] | [1, 1.05] [1.1, 1.1] |  |
| DM2 | [0.7, 0.8] [0.75, 0.85] | [1, 1.05] [1.1, 1.1] | [0.8, 0.9] [0.85, 0.95] | [1.1, 1.2] [1.2, 1.25] | [0.3, 0.4] [0.35, 0.45] | [1.2, 1.35] [1.3, 1.4] | [1.1, 1.2] [1.2, 1.25] |  |
| DM3 | [0.7, 0.8] [0.75, 0.85] | [1.2, 1.35] [1.3, 1.4] | [0.8, 0.9] [0.85, 0.95] | [1, 1.05] [1.1, 1.1] | [0.3, 0.4] [0.35, 0.45] | [1.1, 1.2] [1.2, 1.25] | [0.8, 0.9] [0.85, 0.95] |  |

After assessing the significance of the criteria, it is necessary to average the values obtained in group decision-making. Therefore, the application of IRN Dombi weighted geometric averaging (IRNDWGA) operator for averaging the mentioned values and obtaining an $IRN\left[s_j^r\right]$ matrix is presented below.

For example, the value of C2 criterion (Table 4) according to decision-makers is as follows: $IRN(C_2^{DM1}) = ([0.80, 0.90]; [0.85, 0.95])$, $IRN(C_2^{DM2}) = ([1.00, 1.05]; [1.10, 1.10])$, $IRN(C_2^{DM3}) = ([0.80, 0.90]; [0.85, 0.95])$

When it comes to weight coefficients in RNDWGA, they are $w_{DM} = (0.333, 0.333, 0.333)^T$ since the significance of all DMs is equal. Based on the values shown, Equation (11), and assuming that $\rho = 1$, for the C2 criterion, the aggregation of values is performed:

$$IRNDWGA(C_2) = \begin{cases} C_{l2} = \dfrac{\sum_{j=1}^{3} \varphi_{lj}}{1+\left\{ \sum\limits_{j=1}^{3} w_j \left( \dfrac{1-f\left(\varphi_{lj}\right)}{f\left(\varphi_{lj}\right)} \right)^{\rho} \right\}^{1/\rho}} = \dfrac{2.60}{1+\left(0.333\times\left(\frac{1-0.308}{0.308}\right)+0.333\times\left(\frac{1-0.385}{0.385}\right)+0.333\times\left(\frac{1-0.308}{0.308}\right)\right)} = 0.857 \\[20pt] C_{u2} = \dfrac{\sum_{j=1}^{3} \overline{Lim}(\varphi_j)}{1+\left\{ \sum\limits_{j=1}^{3} w_j \left( \dfrac{1-f\left(\overline{\varphi}_{lj}\right)}{f\left(\overline{\varphi}_{lj}\right)} \right)^{\rho} \right\}^{1/\rho}} = \dfrac{2.85}{1+\left(0.333\times\left(\frac{1-0.316}{0.316}\right)+0.333\times\left(\frac{1-0.368}{0.368}\right)+0.333\times\left(\frac{1-0.316}{0.316}\right)\right)} = 0.945 \\[20pt] C_{l2'} = \dfrac{\sum_{j=1}^{3} \varphi_{lj}}{1+\left\{ \sum\limits_{j=1}^{3} w_j \left( \dfrac{1-f\left(\varphi_{lj}\right)}{f\left(\varphi_{lj}\right)} \right)^{\rho} \right\}^{1/\rho}} = \dfrac{2.80}{1+\left(0.333\times\left(\frac{1-0.304}{0.304}\right)+0.333\times\left(\frac{1-0.393}{0.393}\right)+0.333\times\left(\frac{1-0.304}{0.304}\right)\right)} = 0.920 \\[20pt] C_{u2'} = \dfrac{\sum_{j=1}^{3} \overline{Lim}(\varphi_j)}{1+\left\{ \sum\limits_{j=1}^{3} w_j \left( \dfrac{1-f\left(\overline{\varphi}_{lj}\right)}{f\left(\overline{\varphi}_{lj}\right)} \right)^{\rho} \right\}^{1/\rho}} = \dfrac{3.00}{1+\left(0.333\times\left(\frac{1-0.317}{0.317}\right)+0.333\times\left(\frac{1-0.367}{0.367}\right)+0.333\times\left(\frac{1-0.317}{0.317}\right)\right)} = 0.995 \end{cases}$$

$$= ([0.857, 0.945], [0.920, 0.995])$$

where the value $f(IRN(\varphi_2))$ is obtained as follows:

$$f(IRN(\varphi_2)) = \begin{cases} f\left(Lim(\varphi_2)\right) = \dfrac{Lim(\varphi_1)}{\sum\limits_{i=1}^{3} Lim(\varphi_i)} = \dfrac{0.80}{2.60} = 0.308; \\[16pt] f\left(\overline{Lim}(\varphi_2)\right) = \dfrac{\overline{Lim}(\varphi_i)}{\sum\limits_{i=1}^{3} \overline{Lim}(\varphi_1)} = \dfrac{0.90}{2.85} = 0.316. \\[16pt] f\left(Lim'(\varphi_2)\right) = \dfrac{Lim(\varphi_1)}{\sum\limits_{i=1}^{3} Lim(\varphi_i)} = \dfrac{0.85}{2.80} = 0.304; \\[16pt] f\left(\overline{Lim}'(\varphi_2)\right) = \dfrac{\overline{Lim}(\varphi_i)}{\sum\limits_{i=1}^{3} \overline{Lim}(\varphi_1)} = \dfrac{0.95}{3.00} = 0.317. \end{cases}$$

The other values are averaged in the identical way, and the resulting matrix $IRN\left[s_j^r\right]$ is shown below (Equation (10)).

$$IRN[s_j] = \begin{bmatrix} [0.857, 0.945], [0.920, 0.995] \\ [0.360, 0.462], [0.411, 0.512] \\ [1.300, 1.500], [1.400, 1.550] \\ [0.257, 0.360], [0.309, 0.411] \\ [1.065, 1.145], [1.165, 1.196] \\ [0.529, 0.630], [0.580, 0.680] \\ [1.065, 1.145], [1.165, 1.196] \end{bmatrix}$$

The coefficient $IRN[k_j] = \begin{bmatrix} [1.000, 1.000], [1.000, 1.000] \\ [1.005, 1.080], [1.055, 1.141] \\ \cdots \\ [0.804, 0.835], [0.855, 0.935] \end{bmatrix}$ is obtained using Equation (12) as follows:

$$IRN[k_2] = [2 - 0.955, 2 - 0.920], [2 - 0.945, 2 - 0.857] = [1.005, 1.080], [1.055, 1.143]$$

The coefficient $IRN[q_j] = \begin{bmatrix} [1.000, 1.000], [1.000, 1.000] \\ [0.875, 0.948], [0.926, 0.955] \\ \cdots \\ [0.340, 0.751], [0.579, 1.095] \end{bmatrix}$ is obtained using Equation (13) as follows:

$$IRN[q_2] = \left[ \frac{1.000}{1.143}, \frac{1.000}{1.055} \right], \left[ \frac{1.000}{1.080}, \frac{1.000}{1.105} \right] = [0.875, 0.948], [0.926, 0.995]$$

$$IRN[q_3] = \left[ \frac{0.875}{1.640}, \frac{0.948}{1.538} \right], \left[ \frac{0.936}{1.589}, \frac{0.995}{1.488} \right] = [0.534, 0.616], [0.582, 0.669]$$

The calculation of the relative rough weight of the criterion (Equation (14)):

$$IRN[w_j] = \begin{bmatrix} [0.122, 0.172], [0.147, 0.213] \\ [0.106, 0.163], [0.136, 0.210] \\ \cdots \\ [0.041, 0.129], [0.085, 0.231] \end{bmatrix} \text{ is obtained as follows:}$$

$$IRN[w_1] = \left[ \frac{1.000}{8.225}, \frac{1.000}{5.803} \right], \left[ \frac{1.000}{6.820}, \frac{1.000}{4.733} \right] = [0.122, 0.172], [0.147, 0.211]$$

$$IRN[w_2] = \left[ \frac{0.875}{8.225}, \frac{0.948}{5.803} \right], \left[ \frac{0.926}{6.820}, \frac{0.995}{4.733} \right] = [0.106, 0.163], [0.136, 0.210]$$

Similarly, using Equations (15)–(18), the inverse interval Rough PIPRECIA method is calculated, and the final results of the IRN PIPRECIA method given in Table 5 are obtained.

**Table 5.** Results obtained by applying the IRN PIPRECIA method.

| | IRN PIPRECIA | Inverse IRN PIPRECIA | Final wj | Rank |
|---|---|---|---|---|
| C1 | [0.12, 0.17], [0.15, 0.21] | [0.06, 0.15], [0.12, 0.27] | [0.15, 0.25], [0.20, 0.35] | 2 |
| C2 | [0.11, 0.16], [0.14, 0.21] | [0.06, 0.14], [0.11, 0.25] | [0.14, 0.23], [0.19, 0.33] | 3 |
| C3 | [0.06, 0.11], [0.09, 0.14] | [0.05, 0.09], [0.08, 0.15] | [0.09, 0.15], [0.13, 0.22] | 8 |
| C4 | [0.09, 0.21], [0.14, 0.31] | [0.08, 0.15], [0.13, 0.24] | [0.13, 0.29], [0.21, 0.43] | 1 |
| C5 | [0.05, 0.13], [0.08, 0.20] | [0.08, 0.13], [0.11, 0.19] | [0.09, 0.19], [0.14, 0.29] | 6 |
| C6 | [0.06, 0.15], [0.10, 0.25] | [0.09, 0.15], [0.13, 0.21] | [0.10, 0.23], [0.17, 0.35] | 4 |
| C7 | [0.04, 0.11], [0.07, 0.19] | [0.09, 0.12], [0.11, 0.16] | [0.08, 0.17], [0.12, 0.27] | 7 |
| C8 | [0.04, 0.13], [0.08, 0.23] | [0.11, 0.14], [0.13, 0.18] | [0.10, 0.20], [0.15, 0.32] | 5 |

*4.2. Evaluation of Alternatives Using the Novel IRN EDAS Method*

This section presents an algorithm of the developed IRN EDAS method for the evaluation of road infrastructure locations and the ranking of asphalt production plants. After evaluating the alternatives according to the criteria, it is first necessary to transform the values of individual decision-makers' responses in group decision-making. After transforming the values into interval rough numbers, averaging is also performed using the IRNDWGA operator as explained in the previous section. Table 6 shows the initial interval rough matrix.

**Table 6.** Initial interval rough matrix for the IRN EDAS method.

|     | C1 | C2 | C7 | C8 |
|-----|-----|-----|-----|-----|
| A1  | [6.11, 6.54], [7.00, 7.00] | [5.21, 6.06], [5.47, 6.48] | [5.47, 6.48], [6.11, 6.54] | [4.47, 5.47], [5.43, 5.89] |
| A2  | [6.11, 6.54], [7.00, 7.00] | [5.21, 6.06], [6.11, 6.54] | [5.47, 6.48], [5.82, 6.77] | [4.47, 5.47], [4.79, 6.35] |
| A3  | [7.00, 7.00], [7.00, 7.00] | [5.21, 6.06], [5.47, 6.48] | [5.47, 6.48], [6.43, 6.89] | [4.47, 5.47], [5.21, 6.06] |
| A4  | [3.45, 4.47], [4.47, 5.47] | [4.55, 6.04], [4.55, 6.04] | [5.47, 6.48], [6.43, 6.89] | [5.47, 6.48], [6.11, 6.54] |
| A5  | [3.45, 4.47], [4.11, 4.54] | [4.55, 6.04], [5.47, 6.48] | [5.47, 6.48], [6.43, 6.89] | [5.47, 6.48], [6.11, 6.54] |
| A6  | [6.11, 6.54], [6.43, 6.89] | [4.55, 6.04], [5.21, 6.06] | [5.47, 6.48], [6.43, 6.89] | [5.47, 6.48], [5.47, 6.48] |
| A7  | [5.21, 6.06], [5.47, 6.48] | [6.43, 6.89], [6.43, 6.89] | [6.11, 6.54], [6.43, 6.89] | [4.47, 5.47], [5.21, 6.06] |
| A8  | [2.43, 3.45], [3.10, 3.53] | [6.43, 6.89], [7.00, 7.00] | [6.11, 6.54], [6.43, 6.89] | [4.47, 5.47], [4.80, 5.76] |
| A9  | [5.11, 5.54], [6.00, 6.00] | [6.43, 6.89], [7.00, 7.00] | [6.11, 6.54], [6.11, 6.54] | [4.47, 5.47], [4.47, 5.47] |
| A10 | [2.43, 3.45], [3.42, 3.89] | [1.67, 4.36], [2.56, 4.54] | [5.47, 6.48], [5.82, 6.77] | [4.47, 5.47], [4.79, 6.35] |
| A11 | [6.11, 6.54], [7.00, 7.00] | [1.67, 4.36], [2.82, 5.40] | [2.77, 4.87], [3.83, 5.89] | [4.47, 5.47], [5.21, 6.06] |
| A12 | [1.09, 1.50], [2.10, 2.52] | [1.67, 4.36], [2.61, 5.06] | [6.11, 6.54], [6.43, 6.89] | [4.47, 5.47], [4.79, 6.35] |
| A13 | [3.10, 3.53], [3.45, 4.47] | [2.56, 4.54], [3.53, 5.02] | [6.11, 6.54], [6.43, 6.89] | [4.43, 4.89], [5.11, 5.54] |
| A14 | [7.00, 7.00], [7.00, 7.00] | [2.56, 4.54], [3.53, 5.02] | [6.11, 6.54], [6.43, 6.89] | [4.43, 4.89], [4.80, 5.76] |
| A15 | [6.11, 6.54], [6.43, 6.89] | [3.23, 5.48], [4.04, 6.20] | [6.11, 6.54], [7.00, 7.00] | [4.47, 5.47], [4.47, 5.47] |
| A16 | [6.11, 6.54], [7.00, 7.00] | [6.43, 6.89], [7.00, 7.00] | [6.11, 6.54], [6.43, 6.89] | [4.11, 4.54], [4.20, 5.05] |
| A17 | [7.00, 7.00], [7.00, 7.00] | [6.43, 6.89], [7.00, 7.00] | [6.11, 6.54], [7.00, 7.00] | [4.11, 4.54], [5.00, 5.00] |
| A18 | [6.11, 6.54], [6.11, 6.54] | [6.43, 6.89], [6.43, 6.89] | [6.11, 6.54], [6.43, 6.89] | [4.47, 5.47], [5.21, 6.06] |
| A19 | [4.47, 5.47], [4.79, 6.35] | [5.21, 6.06], [6.11, 6.54] | [6.43, 6.89], [7.00, 7.00] | [4.47, 5.47], [5.43, 5.89] |
| A20 | [3.10, 3.53], [4.11, 4.54] | [5.21, 6.06], [6.11, 6.54] | [6.43, 6.89], [6.43, 6.89] | [4.47, 5.47], [5.47, 6.48] |
| A21 | [1.37, 1.88], [2.40, 2.88] | [5.21, 6.06], [5.21, 6.06] | [6.43, 6.89], [7.00, 7.00] | [4.47, 5.47], [5.47, 6.48] |

Then, in the third step, using Equations (21) and (22), the average solution is calculated forming a matrix:

$$IRN(AV_j) = \begin{bmatrix} [4.71, 5.25], [5.31, 5.71] \\ [4.61, 5.88], [5.22, 6.20] \\ [4.89, 5.34], [5.44, 6.04] \\ [4.42, 5.26], [4.96, 5.73] \\ [4.26, 4.84], [4.83, 5.45] \\ [5.97, 6.41], [6.41, 6.75] \\ [5.79, 6.49], [6.33, 6.82] \\ [4.57, 5.47], [5.12, 5.98] \end{bmatrix}$$

$$\sum_{i=1}^{m} \frac{IRN(x_{ij})}{m} = \begin{bmatrix} \left[ \frac{[6.11+6.11+7+3.45+3.45+6.11+...+1.37]}{21} \right] = 4.71 \\ \left[ \frac{[6.54+6.54+7+4.47+4.47+6.54+...+1.88]}{21} \right] = 5.25 \\ \left[ \frac{[7+7+7+4.47+4.11+6.43+...+2.40]}{21} \right] = 5.31 \\ \left[ \frac{[7+7+7+5.47+4.54+6.89+...+2.88]}{21} \right] = 5.71 \end{bmatrix}$$

After that, a positive distance matrix *IRN(PDAij)* and a negative distance matrix *IRN(NDAij)* in relation to the average solution *IRN(AVj)* are calculated for all criteria. Since this research includes the criteria that belong to the benefit group, Equations (23)–(26) and Equations (29)–(31) are applied. An example of calculation for the element $(pda_{11}) = [0.07, 0.23], [0.34, 0.49]$ is the following:

$$(pda_{11}) = \left[ \frac{0.40}{5.71}, \frac{1.24}{5.31} \right], \left[ \frac{1.76}{5.25}, \frac{2.29}{4.71} \right]$$

$$(B_{11}) = [0.40, 1.24], [1.76, 2.29] = \max(0, [6.11 - 5.71, 6.54 - 5.31], [7 - 5.25, 7 - 4.71])$$

When it comes to the calculation of the element $(nda_{11}) = [0.0, 0.00], [0.00, 0.00]$, it is performed as follows:

$$(nda_{11}) = \left[ \frac{0.00}{5.71}, \frac{0.00}{5.31} \right], \left[ \frac{0.00}{5.25}, \frac{0.00}{4.71} \right]$$

$$(B_{11}) = [0.00, 0.00], [0.00, 0.00] = \max(0, [4.71 - 7, 5.25 - 7], [5.31 - 6.54, 5.71 - 6.11])$$

In the fifth step, the interval rough matrix *IRN(PDAij)* and *IRN(NDAij)* is multiplied with the weight values of the criteria using Equations (32) and (33).

$$(vp_{11}) = [0.01, 0.06], [0.07, 0.17] = [0.07 \times 0.15, 0.23 \times 0.25], [0.34 \times 0.21, 0.49 \times 0.35]$$

$$(vn_{11}) = [0.00, 0.00], [0.00, 0.00] = [0.00 \times 0.15, 0.00 \times 0.25], [0.00 \times 0.21, 0.00 \times 0.35]$$

In the sixth step, the sum of previously weighted rough interval matrices is calculated, using Equations (34) and (35), respectively.

$$sp_1 = \begin{bmatrix} [0.01 + 0 + 0 + 0 + 0 + 0 + 0 + 0 + 0 + 0] = 0.01 \\ [0.06 + 0 + 0 + 0 + 0 + 0 + 0 + 0 + 0 + 0] = 0.06 \\ [0.07 + 0.01 + 0 + 0.03 + 0 + 0 + 0.01 + 0 + 0 + 0] = 0.12 \\ [0.17 + 0.14 + 0.04 + 0.10 + 0 + 0 + 0.03 + 0.09 + 0 + 0] = 0.58 \end{bmatrix}$$

$$sn_1 = \begin{bmatrix} [0 + 0 + 0 + 0 + 0.02 + 0.01 + 0 + 0 + 0 + 0] = 0.03 \\ [0 + 0 + 0 + 0 + 0.04 + 0.03 + 0 + 0 + 0 + 0.01] = 0.108 \\ [0 + 0.03 + 0.01 + 0 + 0.01 + 0.02 + 0 + 0.01 + 0 + 0.01] = 0.09 \\ [0 + 0.07 + 0.05 + 0.13 + 0.14 + 0.13 + 0.06 + 0.11 + 0 + 0] = 0.69 \end{bmatrix}$$

Then, normalization for the matrices from the previous step is performed for all alternatives using Equations (36) and (37), respectively.

$$(nsp_1) = [0.01, 0.44], [0.74, 15.30] = \left[ \frac{0.01}{0.72}, \frac{0.06}{0.13} \right], \left[ \frac{0.12}{0.17}, \frac{0.58}{0.04} \right]$$

$$(nsn_1) = [-9.23, 0.70], [0.61, 0.97] = 1 - \left[ \frac{0.03}{0.89}, \frac{0.08}{0.20} \right], \left[ \frac{0.09}{0.29}, \frac{0.69}{0.07} \right]$$

In the eighth step, the values of all *IRN(ASi)* alternatives are calculated using Equation (38) and they are ranked in descending order, as shown in Table 7.

$$(as_1) = [-4.61, 0.57], [0.68, 8.13] = \left[ \frac{0.01 + (-9.23)}{2}, \frac{0.44 + 0.70}{2} \right], \left[ \frac{0.74 + 0.61}{2}, \frac{15.30 + 0.97}{2} \right]$$

**Table 7.** Final results and ranking obtained using the IRN EDAS method.

| | $IRN(AS_i)$ | Rank |
|---|---|---|
| A1 | [−4.61, 0.57], [0.68, 8.13] | 15 |
| A2 | [−4.58, 0.54], [0.84, 9.48] | 10 |
| A3 | [−6.32, 0.55], [0.58, 9.07] | 17 |
| A4 | [−5.13, 0.39], [0.43, 6.71] | 19 |
| A5 | [−6.10, 0.41], [0.35, 7.62] | 20 |
| A6 | [−2.11, 0.92], [0.69, 9.48] | 2 |
| A7 | [−3.61, 0.95], [0.45, 9.16] | 6 |
| A8 | [−4.62, 0.69], [0.43, 7.31] | 18 |
| A9 | [−4.05, 0.65], [0.46, 7.90] | 13 |
| A10 | [−5.47, 0.75], [0.26, 9.27] | 14 |
| A11 | [−4.92, 1.00], [0.36, 9.99] | 8 |
| A12 | [−6.09, 0.71], [0.13, 9.52] | 16 |
| A13 | [−6.28, 0.37], [0.17, 6.26] | 21 |
| A14 | [−3.88, 0.89], [0.55, 8.33] | 12 |
| A15 | [−2.32, 1.18], [0.63, 9.61] | 1 |
| A16 | [−3.18, 1.04], [0.56, 8.48] | 7 |
| A17 | [−3.16, 1.12], [0.81, 8.54] | 4 |
| A18 | [−3.53, 1.04], [0.43, 9.15] | 5 |
| A19 | [−3.00, 0.49], [0.68, 10.72] | 3 |
| A20 | [−3.93, 0.38], [0.44, 9.44] | 9 |
| A21 | [−4.43, 0.65], [0.40, 9.51] | 11 |

The results presented in Table 7, obtained using the original developed IRN PIPRECIA–IRN EDAS model, show that (A15) Ruma (SP)–Mačvanska Mitrovica–Zasavica has the best characteristics out of the set of locations considered in this study. However, Alternatives A6 and A19 are also variants with remarkably good characteristics since there is very little difference in values compared to the first-ranked alternative. Considering the results obtained and a need to rank the asphalt production plants, it is necessary to group the results from Table 7. The asphalt production plant located on the territory of Subotica is a plant with the best performance regarding the locations considered. It includes locations A16–A18, which are individually positioned in the seventh, fourth and fifth place. The asphalt production plant in Pančevo, which unites the ranks of locations A19–A21, is in second place. The plant (A13–A15) located in Ruma (SP) is third place in the overall ranking, primarily due to A15, which is in first place. However, due to a very bad rank of the A13 location, it is in third place in the overall ranking. The plant in Srbobran (A7–A9) is in fourth place. Very small differences are observed in the ranking of the last and penultimate plants, i.e., asphalt production plants in Bački Petrovac and Rumenka.

## 5. Sensitivity Analysis and Discussion

The comparison of the IRN EDAS model is based on a set of requirements that multi-criteria models should meet in order to adequately model the group decision-making process developed in this paper. The analysis was performed taking into account the following: checking the robustness of the solution through a comparative analysis; validation to supporting GDM; the number of decision making elements.

### 5.1. Checking the Robustness of the Solution in Comparison to Other IRN MCDM Methods

This section of the paper presents the verification of the IRN PIPRECIA–IRN EDAS model through several phases. First, a comparative analysis of the obtained results has been performed with already developed methods: Interval Rough SAW (Simple Additive Weighting method) [29], Interval Rough CoCoSo (Combined Compromise Solution) [20] and Interval Rough COPRAS (COmplex PRoportional ASsessment) [30] methods. The results of the comparative analysis are shown in Figure 2, while the ranks in the comparative analysis are shown in Figure 3.

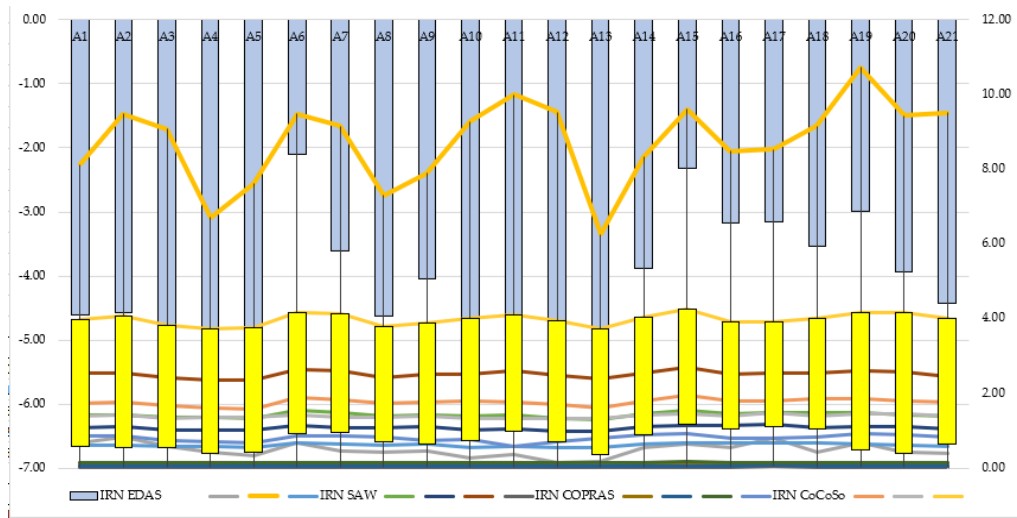

**Figure 2.** Values of alternatives through a comparative analysis.

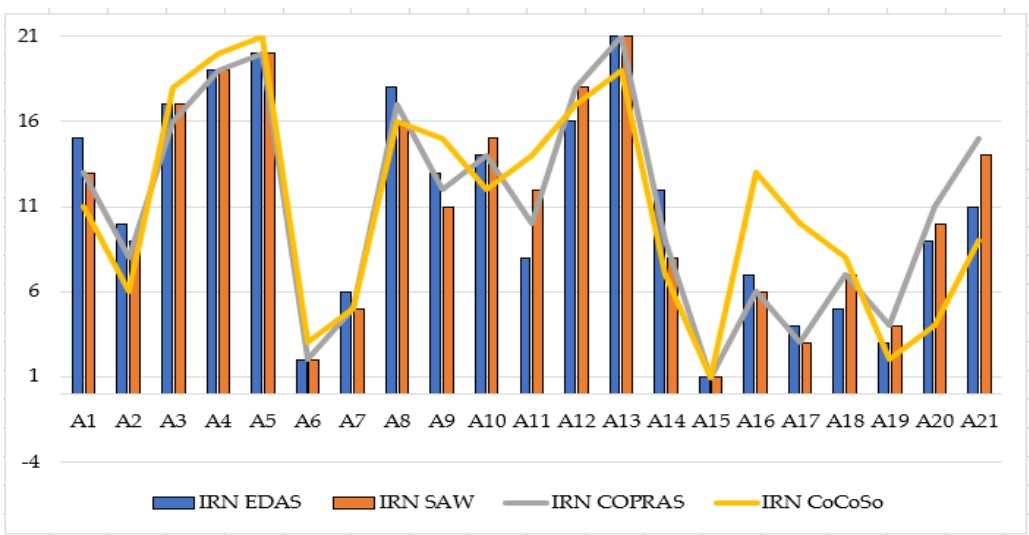

**Figure 3.** Ranks in a comparative analysis with other IRN MCDM methods.

Figure 3 refers to IRN values in a comparative analysis with the mentioned methods. IRN COPRAS has the smallest range, while the interval values in IRN EDAS have the largest range (−6.32–10.72), observing the lower and upper limit of the interval rough number. The interval rough numbers and IRN SAW have a relatively small range (0.54–2.68). The values of interval rough numbers may vary depending on the methodology itself, but the results obtained using the integrated IRN PIPRECIA–IRN EDAS model have been verified, as can be seen from their ranks shown in Figure 3 and the calculated Spearman's coefficient (SCC).

In Figure 3, it can be seen that when comparing the developed methodology with other IRN MCDM methods, there is a change in the ranks of all three methods. This is understandable taking into account the diversity of methodologies and the large number of alternatives being considered. It is important to emphasize that the alternatives that are in the first, second and third place in the original IRN PIPRECIA–IRN EDAS do not change their positions regardless of the applied method. Since changes in the rankings are noticeable, the SCC has been calculated in order to determine the correlation of the obtained results in a comparative analysis. The total SCC is 0.929, which means that the ranks are very highly correlated. When observing a single SCC, it is 0.958 comparing IRN PIPRECIA–IRN EDAS with IRN SAW, 0.966 with IRN COPRAS and 0.864 with IRN CoCoSO.

*5.2. Adequacy to Supporting Group Decision-Making*

GDM contains a great amount of uncertainty and subjectivity, so DMs often have dilemmas when assigning certain values to decision attributes. In this paper, the original PIPRECIA–EDAS model, based on IRNs, is used for processing the uncertainty contained in data in GDM. We suppose that one decision attribute should be assigned a values range from 1 to 5. One DM gave a value between 2 and 3, another DM gave a value between 3 and 4 that should be assigned, while the third DM has no dilemma and assigns a value of 5. In such situations, one of the solutions is to geometrically average. However, the uncertainty that prevailed in a decision-making process would be lost and further calculation would be reduced to crisp values. Also, using fuzzy or grey techniques would entail predicting the existence of uncertainty and subjectively defining the interval by which uncertainty is exploited. Subjectively defined intervals in further data processing can significantly influence the final decision [29,31], which should definitely be avoided if we aim at impartial decision-making.

However, the model based on IRN contains appreciation the uncertainty contained in the data obtained. By applying this methodology, we obtain the values of attributes that fully describe the specified uncertainties without subjectively affecting their values.

The proposed model enables the averaging of decision-makers' preferences in a group decision-making process. Certain studies, such as [32,33], point out that authors should perform averaging using an arithmetic or geometric mean. However, applying such methods of averaging in a group decision-making process, uncertainties and inaccuracies that still exist for decision-makers, i.e., their preferences used for defining an initial matrix, are not considered. Since the proposed IRN PIPRECIA–IRN EDAS model uses interval rough numbers, it is possible to present experts' preferences in such a way as to take into account uncertainties and inaccuracies in group decision-making. The application of IRN reduces the percentage of subjectivity in group decision-making. This indicates the advantages of the proposed IRN PIPRECIA–IRN EDAS model.

Furthermore, advantages and professional contribution of this study can be manifested through measuring performance of asphalt production plants. The best solution can be a benchmark for other asphalt production plants in order to increase their efficiency and achieve better performance.

*5.3. The Number of Alternatives and Criteria*

Most MCDM methods have no restrictions on the size of initial matrix, i.e., the number of potential solutions and criteria used for evaluation. However, it is possible to notice that in many methods, with an increase in the size of initial matrix, some uncertainties and lack of information about some alternatives grow. Since the IRN PIPRECIA–IRN EDAS model has been developed and verified on an example of management in the field of road construction with 21 alternatives and eight criteria considering the uncertainties, an additional advantage of this model is noted.

**6. Conclusions**

In this paper, for the purposes of ranking and selection of asphalt production plants, an original Interval Rough MCDM model has been developed. It has been formed a group decision-making model, which includes the evaluation of a total of 21 locations for road infrastructure maintenance by seven asphalt production plants in the area of Vojvodina on the basis of eight criteria. The advantages of the developed model can be seen in terms of the following facts: robustness of the model determining different phases of sensitivity analysis, and simple procedure of IRN PIPRECIA method after transformation into an interval rough matrix. When it comes to IRN PIPRECIA, the significance of criteria is determined comparing a criterion to the previous one, which means that decision-makers in group decision-making do not have a complex procedure in showing their preferences. The IRN EDAS method defines a negative and positive distance from an average solution, which can allow for more precise decision-making. In addition, the use of interval rough numbers reduces the existence of uncertainty, doubt and subjectivity in group decision-making.

Further, by developing the original interval rough MCDM model, the following contributions can be singled out: (1) Creating a new IRN PIPRECIA–IRN EDAS approach that enables assessment, ranking and selection in the field of road construction in an objective way. (2) Insensitivity of the model increasing the size of initial matrix. Considering that the IRN PIPRECIA–IRN EDAS model has been developed and verified on an example of management in the field of road construction with 21 alternatives and eight criteria considering uncertainties, the advantage of this model can be noted. (3) The model formed enables the assessment of potential solutions regardless of uncertainties that exist in group decision-making. (4) The IRN PIPRECIA–IRN EDAS model enables a flexible group decision-making process and can serve as a useful tool for researchers in the field of road construction management, but also in other different areas of application. (5) The developed model represents an improved MCDM methodology that can serve as a useful tool for managers in a decision-making process.

One of the main limitations of the developed model is the complexity of a mathematical model for the calculation of IRN values. Adding to this, the increase in the number of decision-makers in group decision-making further complicates the application of this

model. However, the limitation can be completely eliminated by creating a software package aimed at user requirements.

This research can help in view of directing future research in terms of scientific and practical application. It is possible to extend the developed model with hybrid fuzzy–rough models, which is one of the directions to be taken in future studies.

**Author Contributions:** Conceptualization, B.M. and M.M.; methodology, Ž.S. and S.S.; validation, D.K.D., and S.J.; investigation, B.M. and M.M.; writing—original draft preparation, B.M. and M.M.; writing—review and editing, D.K.D. and S.S.; supervision, S.J. All authors have read and agreed to the published version of the manuscript.

**Funding:** This research received no external funding.

**Institutional Review Board Statement:** Not applicable.

**Informed Consent Statement:** Not applicable.

**Data Availability Statement:** Not applicable.

**Acknowledgments:** This research (paper) has been supported by the Ministry of Education, Science and Technological Development through the project no. 451-03-68/2020-14/200156: "Innovative scientific and artistic research from the FTS (activity) domain".

**Conflicts of Interest:** The authors declare no conflict of interest.

## Abbreviations

List of Abbreviations:

| Abbreviation | Definition |
| --- | --- |
| IRN | Interval Rough Number |
| MCDM | Multi-Criteria Decision-Making |
| PIPRECIA | PIvot Pairwise RElative Criteria Importance Assessment |
| EDAS | Evaluation based on Distance from Average Solution |
| IRNDWGA | Interval Rough Number Dombi Weighted Geometric averaging Aggregator |
| DEMATEL | Decision-Making Trial And Evaluation Laboratory |
| ANP | Analytical Network Process |
| TOPSIS | Technique for Order Preference by Similarity to Ideal Solution |
| WASPAS | Weighted Aggregate Sum Product Assessment |
| AHP | Analytic Hierarchy Process |
| VIKOR | VIšeKriterijumska Optimizacija i Kompromisno Rešenje |
| HMAR | Hot Mix Asphalt with Reclaimed asphalt pavement |
| HMAW | Hot Mix Asphalt with Warm mix Additive sasobit |
| AE | Almost equal value |
| SM | Slightly more significant |
| MMS | Moderately more significant |
| M | More significant |
| MM | Much more significant |
| DM | Dominantly more significant |
| AM | Absolutely more significant |
| WL | Weakly less significant |
| MLS | Moderately less significant |
| L | Less significant |
| RL | Really less significant |
| ML | Much less significant |
| DL | Dominantly less significant |
| AL | Absolutely less significant |
| DM | Decision-Maker |
| SAW | Simple Additive Weighting method |
| CoCoSo | Combined Compromise Solution |
| COPRAS | COmplex PRoportional ASsessment |
| SCC | Spearman Correlation Coefficient |

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
