# Peer review of "A Novel Integrated Interval Rough MCDM Model for Ranking and Selection of Asphalt Production Plants"

_mathematics, doi:10.3390/math9030269_

Round 1

Reviewer 1 Report

The paper is well written and deals with an interesting topic. However the paper has some shortcoming in its current format. The Authors need to address the following comments and suggestions:

1) In my opinion, given the aim and scope of the journal Mathematics, the main objective of the paper should be to develop a novel interval rough MCDM model. Determining the efficiency of asphalt production plants is only an example of using the new model. Therefore, more important are the principles and advantages of the IRN PIPRECIA – IRN EDAS model  than the asphalt production plants problems.  

2) I suggest the Authors change the title of the paper to: “A novel integrated interval rough MCDM model”.

3) In the abstract, introduction and conclusion sections, the emphasis should be on the novelty and advantages of the new model.

Author Response

Reviewer 1:

Thank you very much for the useful suggestions. We accepted your suggestion and we are sure that this will improve the quality and contribute to a better understanding of the paper.

The paper is well written and deals with an interesting topic. However the paper has some shortcoming in its current format. The Authors need to address the following comments and suggestions:

Comment 1: In my opinion, given the aim and scope of the journal Mathematics, the main objective of the paper should be to develop a novel interval rough MCDM model. Determining the efficiency of asphalt production plants is only an example of using the new model. Therefore, more important are the principles and advantages of the IRN PIPRECIA – IRN EDAS model than the asphalt production plants problems.

Response to comment 1: Yes, you have right. We have shown the advantages of the developed IRN PIPRECIA – IRN EDAS method through the whole paper. This is related to your third comment. In the introduction section, we have shown some important tasks related to road construction that is field of application of the developed model. Also, one part of the introduction section describes just what you asked. Also, in the third section where we have developed integrated IRN MCDM model, we have shown some background of MCDM field and justification for developing such model. Except that, we clearly denote the advantages of the proposed model through sensitivity analysis.

Comment 2: I suggest the Authors change the title of the paper to: “A novel integrated interval rough MCDM model”.

Response to comment 2: Thank you for your suggestion which has sense. However, we have decided that the title of the paper stay the same for the following reasons: - if the title contains the field of exploitation more readers can be interested in our publication, - determining the efficiency of asphalt production plants is an example of using the new model, but not illustrative, than real case study example, - some of the members of the researcher's team belong to the mentioned field and they sure will use and improve this model in future.

Comment 3:  In the abstract, introduction and conclusion sections, the emphasis should be on the novelty and advantages of the new model.

Response to comment 3: Yes, of course. We wrote these sections with mentioned elements. Novelty and contribution are presented in the abstract:

The modern economy needs contemporary models and methods to solve complicated MCDM problems and for these purposes, it has been developed an original Interval Rough Number (IRN) Multi-criteria decision-making (MCDM) model which implies an extension of two methods belonging to the field with interval rough numbers“.

The aim of this paper is to develop a novel interval rough model that can be useful for determining the efficiency of asphalt production plants“.

in introduction section:

In addition to the professional contribution that is reflected in determining the efficiency through the ranking of asphalt production plants that can have an impact on the environment, but also the functioning of the economic system, the following scientific contributions are highlighted. Techniques belonging to the field of MCDM, especially in terms of integration with rough and interval rough numbers [7,8], represent an extremely powerful and useful approach for more precise decision-making in a group decision-making process. An extension of the PIPRECIA and EDAS methods with interval rough numbers has been performed, which is a great contribution of this study (IRN PIPRECIA – IRN EDAS model). A detailed algorithm of both developed approaches is presented in Section 3. The contribution can be seen through the development of an entire interval rough MCDM model, implying an extension of the PIPRECIA method, which determines the significance of the criteria, and the EDAS method, which is used to rank potential solutions. The new Interval Rough EDAS approach is used to assess the efficiency of asphalt plants through the locations of the roads they serve, which enables the reduction of subjectivity and imprecisions that occur on a daily basis when making decisions. This model provides an objective aggregation of experts' decisions with full respect for inaccuracy and subjectivity that prevails in group decision-making. The development of these models further enhances the literature that considers the theoretical and practical application of multi-criteria techniques.

in conclusion section:

The advantages of the developed model can be seen in terms of the following facts: robustness of the model determining different phases of sensitivity analysis, simple procedure of IRN PIPRECIA method after transformation into an interval rough matrix. When it comes to IRN PIPRECIA, the significance of criteria is determined comparing a criterion to the previous one, which means that decision-makers in group decision-making do not have a complex procedure in showing their preferences. The IRN EDAS method defines a negative and positive distance from an average solution, which can allow for more precise decision-making. In addition, the use of interval rough numbers reduces the existence of uncertainty, doubt and subjectivity in group decision-making.

Further, by developing the original interval rough MCDM model, the following contributions can be singled out: 1) Creating a new IRN PIPRECIA – IRN EDAS approach which enables assessment, ranking and selection in the field of road construction in an objective way. 2) Insensitivity of the model increasing the size of initial matrix. Considering that the IRN PIPRECIA – IRN EDAS model has been developed and verified on an example of management in the field of road construction with 21 alternatives and eight criteria considering uncertainties, the advantage of this model can be noticed. 3) The model formed enables the assessment of potential solutions regardless of uncertainties that exist in group decision-making. 4) The IRN PIPRECIA – IRN EDAS model enables a flexible group decision-making process and can serve as a useful tool for researchers in the field of road construction management, but also in other different areas of application. 5) The developed model represents an improved MCDM methodology that can serve as a useful tool for managers in a decision-making process.

Reviewer 2 Report

The paper provides an interesting read about multi-criteria decision making for ranking the selection of asphalt production plants. There are some few minor points that should be addressed.

1) The abstract, line 28 must include the implication of the findings on selecting asphalt plants. This implication forms the distinguishing basis of the paper.

2) From line 327 to 339, the formulas should be numbers for the purpose of consistency. 

3) Add more discussions to section 5.2. This is where you can discuss the implications of findings on production activities and future research. 

4) Revise the reference list in accordance with the journal requirement. The reference list at the end of the file is not in the right format. 

Author Response

Reviewer 2:

Thank you very much for the useful suggestions. We accepted your suggestion and we are sure that this will improve the quality and contribute to a better understanding of the paper.

The paper provides an interesting read about multi-criteria decision making for ranking the selection of asphalt production plants. There are some few minor points that should be addressed.

Comment 1: The abstract, line 28 must include the implication of the findings on selecting asphalt plants. This implication forms the distinguishing basis of the paper.

Response to comment 1: We have added the following: „The obtained results show that (A15) Ruma (SP) – Mačvanska Mitrovica – Zasavica has the best characteristics out of the set of locations considered in this study. However, Alternatives A6 and A19 are also variants with remarkably good characteristics since there is very little difference in values compared to the first-ranked alternative.“

Comment 2: From line 327 to 339, the formulas should be numbers for the purpose of consistency.

Response to comment 2: Thank you for your remark. We have adopted your suggestion.

Comment 3: Add more discussions to section 5.2. This is where you can discuss the implications of findings on production activities and future research. 

Response to comment 3: Thank you for your suggestion. We have extended this section. Please see page 18.

Comment 4: Revise the reference list in accordance with the journal requirement. The reference list at the end of the file is not in the right format.

Response to comment 4: We have corrected the whole list of references.

Reviewer 3 Report

This paper develops a novel interval rough model that can be useful for determining the efficiency of asphalt production plants. Its content is sound and clear. Writing is fair. Could be of interest to relevant researchers. Comparative results show good performance of the proposed method. It is suggested for publication. Two minor questions that need to be paid attention to. Why is interval rough numbers superior to interval arithmetic? There are so many abbreviations that quite distracting the readers. Please provide a dedicated section for terminology. And also try to reduce unnecessary ones.

Author Response

Reviewer 3:

Thank you very much for the useful suggestions. We accepted your suggestion and we are sure that this will improve the quality and contribute to a better understanding of the paper.

Comment 1: This paper develops a novel interval rough model that can be useful for determining the efficiency of asphalt production plants. Its content is sound and clear. Writing is fair. Could be of interest to relevant researchers. Comparative results show good performance of the proposed method. It is suggested for publication. Two minor questions that need to be paid attention to. Why is interval rough numbers superior to interval arithmetic? There are so many abbreviations that quite distracting the readers. Please provide a dedicated section for terminology. And also try to reduce unnecessary ones.

Response to comment 1: Thank you for your remarks.

The first task have explained at the beginning of 5.2 section. „Group decision-making contains a great amount of uncertainty and subjectivity, so DMs often have dilemmas when assigning certain values to decision attributes. In this paper, the original PIPRECIA - EDAS model, based on interval rough numbers (IRN), is applied to process the uncertainty contained in data in group decision-making. If we suppose that one decision attribute should be assigned a value presented by a qualitative scale whose values range from 1 to 5. One  DM may consider that the decision attribute should have a value between 2 and 3, another DM may consider that a value between 3 and 4 should be assigned, while the third DM has no dilemma about the value of the decision attribute and assigns a value of 5. In such situations, one of the solutions is to geometrically average two values between which individual decision-makers are in doubt. However, in such situations, the uncertainty that prevailed in a decision-making process would be lost and further calculation would be reduced to crisp values. On the other hand, the use of fuzzy or grey techniques would entail predicting the existence of uncertainty and subjectively defining the interval by which uncertainty is exploited. Subjectively defined intervals in further data processing can significantly influence the final decision [29,31], which should definitely be avoided if we aim at impartial decision-making.

On the contrary, model based on IRN includes exploiting the uncertainty that exists in the data obtained. By applying the arithmetic operations that are previously explained, we obtain the values of attributes that fully describe the specified uncertainties without subjectively affecting their values.“

The second task: We have created list of abbreviations in Appendix A1.

Round 2

Reviewer 1 Report

The Authors improved the paper.